# A Clinico-Genetic Score Incorporating Disease-Free Intervals and Chromosome 8q Copy Numbers: A Novel Prognostic Marker for Recurrence and Survival Following Liver Resection in Patients with Liver Metastases of Uveal Melanoma

**DOI:** 10.3390/cancers16193407

**Published:** 2024-10-07

**Authors:** Pascale Mariani, Gaëlle Pierron, Khadija Ait Rais, Toufik Bouhadiba, Manuel Rodrigues, Denis Malaise, Livia Lumbroso-Le Rouic, Raymond Barnhill, Marc-Henri Stern, Vincent Servois, Toulsie Ramtohul

**Affiliations:** 1Department of Surgical Oncology, Institut Curie, PSL Research University, 75005 Paris, France; mohammedtoufik.bouhadiba@curie.fr; 2Department of Genetics, Somatic Genetic Unit, Institut Curie, PSL Research University, 75005 Paris, France; gaelle.pierron@curie.fr (G.P.); khadija.aitrais@curie.fr (K.A.R.); 3Department of Medical Oncology, Institut Curie, PSL Research University, 75005 Paris, France; manuel.rodrigues@curie.fr; 4INSERM U830, DNA Repair and Uveal Melanoma (D.R.U.M.), 75005 Paris, France; marc-henri.stern@curie.fr; 5Department of Ocular Oncology, Institut Curie, PSL Research University, 75005 Paris, France; denis.malaise@curie.fr (D.M.); livia.lumbroso@curie.fr (L.L.-L.R.); 6Inserm U1288, Institut Curie, PSL Research University, 91400 Orsay, France; 7Department of Translational Research, Institut Curie, 75005 Paris, France; raymond.barnhill@curie.fr; 8Department of Radiology, Institut Curie, PSL Research University, 75005 Paris, France; vincent.servois@curie.fr (V.S.); toulsie.ramtohul@curie.fr (T.R.)

**Keywords:** genetic profile, prognostic value, uveal melanoma, liver metastasis resection

## Abstract

**Simple Summary:**

In a retrospective study of 86 patients, we identified independent predictors of recurrence-free survival (RFS) and overall survival (OS) after the resection of liver metastases of uveal melanoma using a multivariable Cox model. A disease-free interval of ≤24 months and a chromosome 8q surgain were associated with worse survival. With these two parameters, we built a novel clinico-genetic score that defined three risk groups with distinct prognoses. This novel score identified patients with a high risk of relapse after surgery. These patients may benefit from neoadjuvant or adjuvant systemic therapy following complete surgical resection with the hope of improving survival outcomes.

**Abstract:**

Surgical treatment of liver metastases of uveal melanoma (LMUM) could be proposed for selected patients. This retrospective study examined the prognostic significance of the genetic profiles of liver metastases after LMUM resection. A total of 86 patients treated with resection for LMUM, who underwent genetic analysis of liver metastasis, were included. A multivariable Cox model identified the independent predictors of recurrence-free survival (RFS) and overall survival (OS). The disease-free interval (DFI) and a chromosome 8q surgain (>3 copies) were independent predictors and categorized patients into three risk groups with distinct postoperative prognoses. For the low-, intermediate-, and high-risk scores of recurrence, the median RFS values were 15 months (95% CI: 10–22), 6 months (95% CI: 4–11), and 4 months (95% CI: 2–7), and the median OS values were 86 months (95% CI: 55-NR), 25 months (95% CI: 17–48), and 19 months (95% CI: 12–22), respectively. The predictive accuracy of this scoring system was demonstrated by a mean area under the curve (AUC(t)) of 0.77 (95% CI: 0.65–0.90) for RFS and 0.81 (95% CI: 0.70–0.92) for OS. This novel score, based on a DFI of ≤24 months combined with a chromosome 8q surgain, identifies patients at a high risk of early recurrence and could help clinicians to propose perioperative treatment.

## 1. Introduction

Liver metastases are the leading cause of death in patients with uveal melanoma (UM). Despite effective treatment of ocular UM, between 30 and 50% of patients will usually present isolated liver metastases with a median overall survival (mOS) of approximately 1 year [1,2,3]. The mOS increased to 21.7 months with the newly developed bispecific fusion protein TEBENTAFUSP in HLA-A*02:01 patients only [4]. Even with this new treatment, patients’ prognosis remains poor. Due to the predominant liver involvement, the National Comprehensive Cancer Network guidelines have recommended some liver-directed therapies for those patients. Among them, locoregional perfusion delivering high-dose chemotherapeutics can be carried out using two approaches: intrahepatic perfusion (IHP), a surgical procedure; or percutaneous hepatic perfusion (PHP), a radiological procedure [5]. In the most recent studies, the mOS values for locoregional perfusion of the liver were 17.1 months for IHP and 20.5 months for PHP [6,7,8]. However, questions remain for resectable oligometastatic patients for whom the surgical resection of liver metastasis of uveal melanoma (LMUM) remains associated with the best survival rates for very select patients with low tumor burdens in the liver [9,10]. In our experience, patients with microscopically complete resections achieved a mOS of 27 months [11]. The prognostic factors of surgical treatment with curative intent include the performance status (PS); the lactate dehydrogenase (LDH) level; the disease-free interval (DFI), defined as the time from the treatment of UM to the diagnosis of LMUM; and the tumor burden visible in liver imaging [12]. These preoperative prognostic factors do not fully capture the progression of patients in terms of recurrence-free survival (RFS) and overall survival (OS).

After treatment of the ocular UM, the metastatic risk depends on the UM AJCC TNM classification and genetic analysis of the primary tumor [13]. However, the prognostic significance of the genetic characteristics of metastases has not been extensively studied. A recent review of ten studies, each including over five patients, found none that conducted a comprehensive analysis of the genetic characteristics of metastases, including the chromosome copy number variation (CNV) in chromosomes 3 and 8 and the driver mutations (*GNAQ*, *GNA11*, *CYSLTR2*, and *PLCB4*) and secondary somatic mutations (*BAP1*, *SF3B1*, and *EIF1AX* = BSE mutations) associated with this disease [14]. Regarding patient survival specific to genetic anomalies in metastases, only four studies have reported findings on the genetic abnormalities of metastases [12,14]. Only one study, including 11 patients, showed that genetic analysis of metastases could predict patient survival [15].

This study aimed to explore whether the genetic profiles of metastases, alongside established prognostic factors, could predict patient survival in patients who had surgical resection of LMUM with curative intent and for whom genetic analysis of metastases was performed.

## 2. Materials and Methods

Following international guidelines [16], all patients were screened for liver metastasis every 6 months. Liver MRI was the screening modality for high-risk patients (exhibiting either the AJCC ocular tumor T3/4 categories or the presence of monosomy of chromosome 3). Liver ultrasound was the chosen modality for other patients. This retrospective study was approved by the Institutional Ethics Review Board. Institutional written informed consent, including consent for somatic genetic analysis, was obtained from all patients (IC-MU-CHIR02-20). We identified patients with LMUM, treated with the intention of curative surgical resection, with our maintained prospective database. The inclusion criteria were histologically confirmed LMUM, resectable liver metastasis in cross-sectional imaging, and genetic analysis of LMUM. Patients with extrahepatic disease were excluded.

Clinical and imaging data were recorded at liver metastasis diagnosis. Capsular and parenchymal miliary disease were identified by exhibiting more than one lesion inferior to 5 mm in size. A major hepatectomy was defined as the resection of more than three segments. Metastases were classified according to two classifications used to establish the prognosis of primary ocular tumors based on chromosome copy number variations, including the status of chromosome 3 and chromosome 8 without (Cassoux) or with (TCGA) the identification of an 8q surgain (defined as strictly more than 3 copies). The Cassoux classification was defined as follows: low risk—disomy 3 (D3)/8 normal (8nl); intermediate risk—monosomy 3 (M3)/8 nl or D3/8q gain (8g); or high risk—M3/8g [17]. The TCGA classification was defined as follows: low risk—D3/8nl; intermediate risk—D3/8g; high risk—M3/8g; or very high risk—M3/8g with an 8q surgain (strictly more than 3 copies). M3/8nl alterations were classified as being of intermediate risk [18].

This study complied with the tenets of the Declaration of Helsinki and was written per the REMARK criteria [19].

DNA from snap-frozen tumor samples was extracted, qualified, and quantified as described in [20]. Comparative genomic hybridization (array-CGH) assessed the copy number variations (CNVs) in both resected liver metastases and frozen primary tumors (51 patients) using the NimbleGen or Agilent technologies [20]. Single-nucleotide variations (SNVs) were identified via sequencing using NGS panels, either PUMA (panel for the detection of uveal melanoma alterations) or DRAGON (detection of relevant alterations in genes involved in oncogenetics), followed by bioinformatics analysis [21,22]. Two expert geneticists reviewed the samples (K.A.R. and G.P.).

Continuous variables were analyzed using the Wilcoxon–Mann–Whitney and Student’s tests according to the distribution normality. Categorical variables were analyzed using the χ2 test or Fisher’s exact test. The disease-free interval (DFI) was defined as the time from ocular tumor diagnosis to liver metastasis diagnosis via imaging. The DFI was categorized as ≤24 or >24 months according to two previous publications [12,23]. Recurrence-free survival (RFS) was defined as the time from liver resection to radiologically defined liver relapse or death due to any cause. Overall survival (OS) was defined as the time from liver resection to death from metastatic evolution. The relationship between genetic data and prognosis was established via Cox proportional-hazards univariable regression for variables associated with RFS and OS at a significance level of *p* < 0.05 and with less than 10% missing data and then applying a backward stepwise approach to retain significant factors in the final model. Three Cox multivariable models were assessed to test the prognostic value of 8q surgain in metastatic patients undergoing surgery, each exploring different genetic classifications: the Cassoux classification (model 1 [17]) without information about 8q surgain, and the TCGA classification (model 2 [18]) with information about 8q surgain and 8q surgain alone without the status of chromosome 3 (model 3). Schoenfeld residuals were used to check the proportional hazard assumption, which was present. All analyses were performed using the SAS software (version 9.4, SAS Institute, Cary, NC, USA). All statistical tests were two-sided. *p*-values of less than 0.05 were considered statistically significant.

## 3. Results

Between January 2008 and December 2018, 627 out of 3659 UM patients developed LMUM. Moreover, 86 patients with available genetic profiles—including CNVs, driver mutations, and BSE mutations in their resected liver metastases—were eligible for this study.

The median disease-free interval (DFI) from ocular UM to LMUM was 38 months (interquartile range [IQR]: 17–71).

Postoperative morbidity was reported according to the Clavien–Dindo classification [24]. Within 30 days’ post-surgery, 5 patients had grade 1–2 complications, and 4 had grade-3–4 complications. By 90 days, 1 patient had a grade-1 complication, and another had a grade-3 complication. No postoperative mortality was reported within 90 days. A total of 80 patients had a macroscopically complete resection, including 71 without microscopic involvement of the margins (R0) and 9 with microscopic involvement (R1). Meanwhile, 6 patients had an incomplete resection with unresected macroscopic liver metastases (R2). R2 patients were excluded from the RFS analysis.

The baseline characteristics of patients with or without chromosome 8q surgain were largely similar, except for the DFI, the LDH level, the presence of capsular miliary disease, the histological features of metastases, the presence of *SF3B1* and *BAP1* mutations, and the Cassoux and TCGA metastatic risk classification classes (Table 1 and Table 2).

### 3.1. Independent Prognostic Factors Associated with RFS and OS

The median follow-up after surgical treatment was 75 months (95% CI: 60–88 months).

The RFS and OS had a statistically significant association (hazard ratio [HR] = 0.90; 95% IC: 0.87–0.94; *p* < 0.001) (Appendix A).

The median RFS (mRFS) was 10 months (IQR: 6–11) (Appendix A). In the univariable analysis (Appendix A), a patient age of >50 years, the presence of a ciliary body extension, a DFI of ≤24 months, the presence of capsular miliary disease, the presence of a *BAP1* mutation, TCGA classification (very-high-risk versus high-risk classes), and the presence of a chromosome 8q surgain were associated with a lower RFS. In the multivariable analyses (Table 3), only a DFI of ≤24 months and the presence of a chromosome 8q surgain remained independent factors associated with a shorter RFS.

The mOS was 31 months (IQR: 23–49) (Appendix A), and it was 35 months (95% CI: 23–55) and 21 months (95% CI: 4–42) after macroscopically complete resection (R0/R1) and incomplete resection (R2) (Appendix A), respectively. In the univariate analysis (Appendix A), a patient age of >50 years, a DFI of ≤24 months, the presence of a capsular miliary disease, incomplete surgical resection, the presence of a *BAP1* mutation, the Cassoux risk classification (high versus intermediate), the TCGA risk classification (very high versus high), and chromosome 8q surgain were associated with a lower OS. In the multivariate analyses (Table 4), a DFI of ≤24 months and the presence of chromosome 8q surgain remained independent factors associated with a lower OS.

The comparison of the three models including genetic characteristics—the Cassoux classification, TCGA classification, and chromosome 8q surgain—for predicting RFS and OS is detailed in Appendix A. Model 3 (DFI of ≤24 months and chromosome 8q surgain) was the most effective, with a mean AUC(t) of 0.77 (95% CI: 0.65–0.90) for RFS and a mean AUC(t) of 0.81 (95% CI: 0.70–0.92) for OS (Appendix A). Consequently, we adopted this model to build a preoperative clinico-genetic score.

### 3.2. Preoperative Clinico-Genetic Risk Score

We defined a clinico-genetic risk score to predict postoperative survival: low score (= 0), DFI of >24 months and no chromosome 8q surgain; intermediate score (= 1), DFI of ≤24 months or chromosome 8q surgain; and high score (= 2), DFI of ≤24 months and chromosome 8q surgain.

This categorization resulted in three distinct groups with significantly different RFS and OS values (*p* < 0.001) (Figure 1).

The mRFS values for the low-, intermediate-, and high-risk scores were 15 months (95% CI: 10–22), 6 months (95% CI: 4–11), and 4 months (95% CI: 2–7), respectively. Similarly, the mOS values for these groups were 86 months (95% CI: 55-NR), 25 months (95% CI: 17–48), and 19 months (95% CI: 12–22), respectively. Moreover, the median DFI significantly varied across the risk groups, recorded at 72, 30, and 14 months for the low-, intermediate-, and high-risk groups (IQRs: 52–108, 19–45, and 9–20), respectively.

### 3.3. Comparison of CNVs and Mutations between UM and Resected Liver Metastases

The CNVs of the 86 liver metastases according to their chromosome 8q status are illustrated in Figure 2.

In terms of the chromosome CNVs, the concordance rates among the 51 analyzable pairs according to the Cassoux and TCGA classifications, and a classification based solely on the chromosome 8q copy number, were 96% (49/51), 82% (42/51), and 82% (42/51), respectively.

Regarding the TCGA classification and the classification focusing on the chromosome 8q copy number alone, 18% (9/51) of the pairs experienced a risk class shift due to an additional increase in the chromosome 8q copy number in the metastases. For the Cassoux classification, which did not consider the specific number of chromosome 8q copies, no change was observed between the UM and the LMUM.

Regarding the driver mutations, for the 48 analyzable pairs (3 non-contributive pairs), we found 24/48 (50%) GNAQ, 22/48 (45.8%) GNA11, 1/48 (2%) CYSLTR2, and 1/48 (2%) PLCB4. The concordance between ocular tumors and liver metastases was 100%.

Regarding BSE mutations, we found 36/48 (75%) BAP1, 10/48 (20.8%) SF3B1, and 2/48 (2%) EIF1AX mutations for the 48 analyzable pairs. The concordance between ocular tumors and liver metastases was 100%. In these concordant cases, one patient presented a nonexclusive mutation of BAP1 and SF3B1.

## 4. Discussion

This retrospective study is the first to report the adverse prognostic impact of chromosome 8q surgain in patients with LMUM, particularly when associated with a disease-free interval (DFI) of 24 months or less, in those undergoing surgical resection of their liver metastases.

In this study, 93% (80 out of 86) of the patients treated with curative intent had a complete surgical resection, indicating a highly selective cohort of patients with oligometastatic disease. This surgical outcome is attributed, in part, to the systematic use of preoperative liver MRI, which allows the optimal mapping of the lesions, completed with intraoperative liver ultrasound for a comprehensive liver examination.

Compared with that in our previous publication [11], the mOS for patients undergoing macroscopically complete resection (which combined R0 and R1) was improved by 8 months (35 months in this study vs. 27 months post-R0 resection in 2009). This result is among the best mOS values published after the treatment of mUM patients. It comforts us that surgical treatment remained a therapeutic option in patients who underwent R0 resection. If we compare this result with other liver-directed therapies especially those with percutaneous hepatic perfusion (PHP) with melphalan/HDS, in recently published studies, the mOS was only 20.5 months for 91 unresectable patients with liver tumor invasion ranging from 25 to 50%. The main difference was that patients could receive multiple treatments with this minimally invasive technique [8]. The mOS post-R0 surgical treatment was also superior to medical treatments including first-line tebentafusp (with an mOS of 21.7 months in a randomized phase 3 study in previously untreated mUM) for a population with a greater tumor burden [4]. In this study, the mRFS did not significantly differ between R0 and R1 resections regarding the quality of surgical resection (Appendix A). Our results agree with Trivedi et al.’s recent publication [10].

Patients were monitored via liver MRI every 3 months after surgical treatment. They did not receive any postoperative treatment if the resection was complete. Upon recurrence, either the patient was eligible for a new local treatment or they received medical treatment. This increase in OS in this study may also have been due to the broader range of treatments available postoperatively today compared with 2009 [11], with half of our patients receiving at least three lines of treatments following surgical relapse (Figure 3).

While RFS was associated with OS in this study, the primary concern in the surgical treatment of LMUM was RFS. Our results show a median RFS of 10 months, aligning with the recent literature [10]. Given that patients received no preoperative or postoperative treatment before recurrence, the RFS reflects the natural progression of the metastatic disease. Identifying clinical and biological factors of “aggressiveness” of the metastatic disease is crucial for enhancing patient management. Our mRFS of 10 months is equivalent to the FOCUS study’s PFS of 9 months despite a response rate of 36.3%. PHP aims to treat macro- and micrometastatic liver disease, whereas surgical resection only treats detectable metastases [8]. This supports our hypothesis that factors other than the usual clinical ones must be considered when treating these patients.

### 4.1. DFI

The DFI was the only clinical factor integrating our clinico-genetic score. According to international guidelines, the DFI relies on biannual liver imaging, including ultrasound for low-risk patients and liver MRI for high-risk patients [16]. Regarding the liver MRI screening results, whether carried out in our center or outside, MRI results were systematically reanalyzed by two expert radiologists during a weekly multidisciplinary uveal melanoma meeting. Moreover, the DFI found as a prognostic factor in this study agrees with several previous publications and exhibits the same cutoff value [11,12,23]. Some authors expressed reservations that there may be inaccuracy in measuring the DFI depending on the rhythm of imaging screening and the imaging method used [14]. We believe that our current institutional organization, based on liver MRI, minimizes this risk.

### 4.2. Chromosome 8q Surgain

Chromosome 8q surgain corroborates the TCGA classification for primary UM based on the chromosome 8q copy number. This finding suggests that a tumor’s metastatic potential persists in the liver, despite rare discrepancies in specific chromosomal conditions. Nevertheless, the TCGA classification does not consider cases with disomy of chromosome 3 associated with an 8q surgain, which was reported in 4/66 cases of UM in [25]. Although rare, this chromosomal condition was present in three metastatic patients in our series.

The prognostic value of the genetic characteristics of LMUM was highlighted in a recent review [14]. Specifically, this study is pioneering in its focused examination of the chromosome 8q copy number as a prognostic marker in patients with metastatic disease. LMUM has more oncogenic alterations than primary UM, notably an increased chromosome 8q copy number. A previous study of 25 patients revealed that a gain in chromosome 8q was observed in 96% (24 out of 25) of the metastases examined. Additionally, in a subset analysis of 13 matched pairs of primary ocular tumors and their corresponding metastases, it was discovered that 23% of the cases had a higher number of 8q copies in the metastases [26]. Similarly, another study involving 35 patients demonstrated that the metastases harbored more copies of chromosome 8q than the primary tumors (six versus four; *p* = 0.002) [27]. Our findings closely align with these observations, revealing an increase in chromosome 8q copy numbers in 18% of liver metastases relative to the primary UM. No prognostic value for *GNAQ* and *GNA11* driver mutations was identified within our oligometastatic cohort, which significantly differed from Terai’s study population [28]. Similarly, no prognostic value was observed for secondary BSE somatic mutations, including *BAP1* mutations. The available data on the BAP1 mutation are mixed, particularly concerning patients undergoing medical treatment. One study of 89 patients found that the BAP1 mutation was associated with shorter progression-free survival, yet it did not evaluate overall survival [29]. Conversely, Terai’s research with 87 patients linked the BAP1 mutation to overall survival in a univariate analysis but not in a multivariable context [28]. Consequently, our findings do not support a prognostic role for the commonly described driver or secondary somatic mutations. This underscores the potential relevance of other secondary genetic or epigenetic changes related to chromosome 8q in influencing patient outcomes, as suggested by Ehlers et al. [30], who showed that overexpression of the DDEF1 gene located at 8q24 increases cell motility and may act as an oncogene in this cancer.

### 4.3. Scientific Rationale for Our Clinico-Genetic Score

Our novel scoring system enabled us to categorize patients into three distinct groups, each demonstrating significantly different survival outcomes. The genetic characteristics of primary uveal melanoma (UM) influence the latency period between the initial treatment and the appearance of metastases, which can vary from several months to decades [31]. The DFI reflects the genetic alterations in the primary tumor, such as monosomy 3 and *BAP1* mutation, which lead to the earlier onset of metastases via a shorter liver intrahepatic metastatic dormancy [32]. This phenomenon might recur post-surgery for hepatic micrometastases that are undetectable either before or during the operation, as indicated by the brief recurrence-free survival (RFS) of less than 10 months in patients with a DFI of 24 months or less. In addition, we confirmed that the chromosome 8q copy number increases in the metastatic stage compared with primary UM. The gain in chromosome 8q is recognized as an adverse prognostic indicator in various hepatic or extrahepatic cancers [33,34]. Given the high prevalence of 8q surgain in LMUM in our cohort (46.5%), we, along with other researchers, propose that the 8q region harbors genes that might drive metastasis progression [35].

Given our results, we believe that genomic analysis of metastases, including the chromosome 8q copy number, should be considered when stratifying patients receiving liver-directed treatment for uveal melanoma metastases to better identify patients with a potential for rapid progression after local treatment and better adapt their subsequent management.

### 4.4. Limitations

This study had limitations: (1) This was a single-center retrospective study based on a series of patients selected for the availability of genetic data. (2) The patients included had a limited extent of liver metastatic disease. (3) Biological data of prognostic interest, such as ctDNA, were unavailable for these patients, even though preoperative ctDNA has emerged as an important parameter [36]. (4) Integrating genetic information into treatment decisions requires genetic analysis from the UM, obtained either via enucleation or fine-needle aspiration biopsy. However, according to our findings, we acknowledge an 18% discrepancy related solely to the chromosome 8q copy number, whether from liver metastasis biopsies or the primary tumor. (5) Furthermore, we did not study additional mutations specific to the metastases, which could potentially alter patient prognoses. This aspect will be addressed in future research.

## 5. Conclusions

In conclusion, this study is pioneering in demonstrating that a combination of a DFI of less than 24 months and chromosome 8q gain in LMUM identifies a group of patients at high risk of relapse. These patients may benefit from neoadjuvant or adjuvant systemic therapy following complete surgical resection, with the hope of improving survival outcomes.

## Figures and Tables

**Figure 1 cancers-16-03407-f001:**
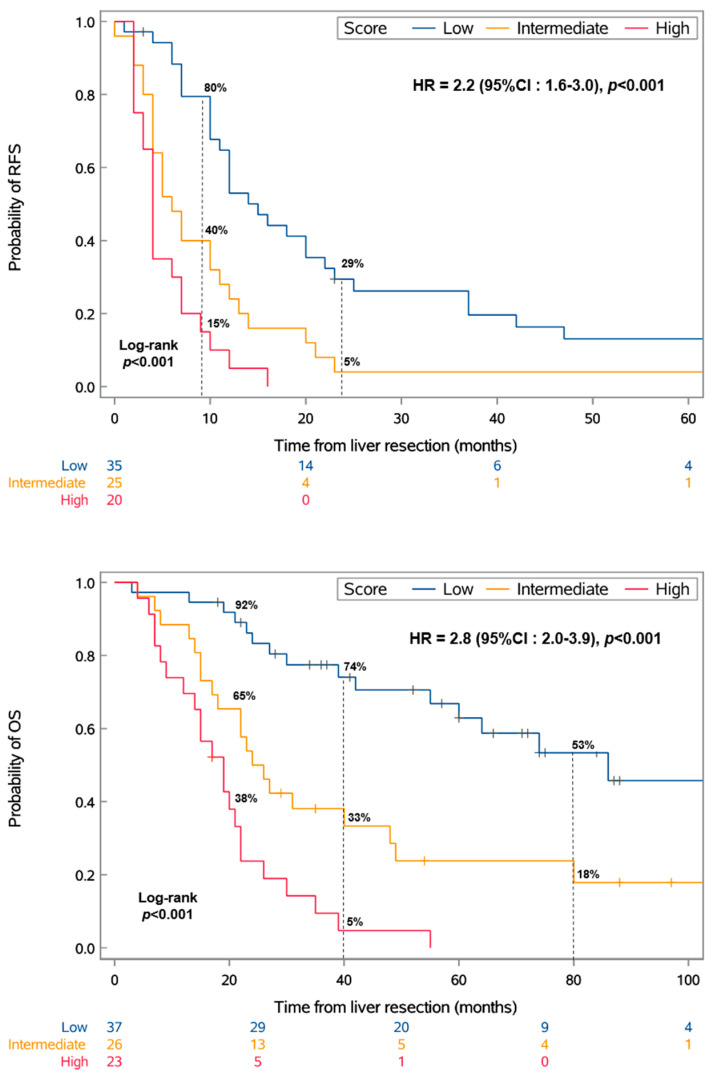
Kaplan–Meier analysis of recurrence-free survival (RFS) and overall survival (OS) by the score. The score was defined as follows: low, DFI of >24 months and no chromosome 8q surgain; intermediate, either a DFI of ≤24 months or chromosome 8q surgain; or high, DFI of ≤24 months and chromosome 8q surgain. HR: hazard ratio; CI: confidence interval.

**Figure 2 cancers-16-03407-f002:**
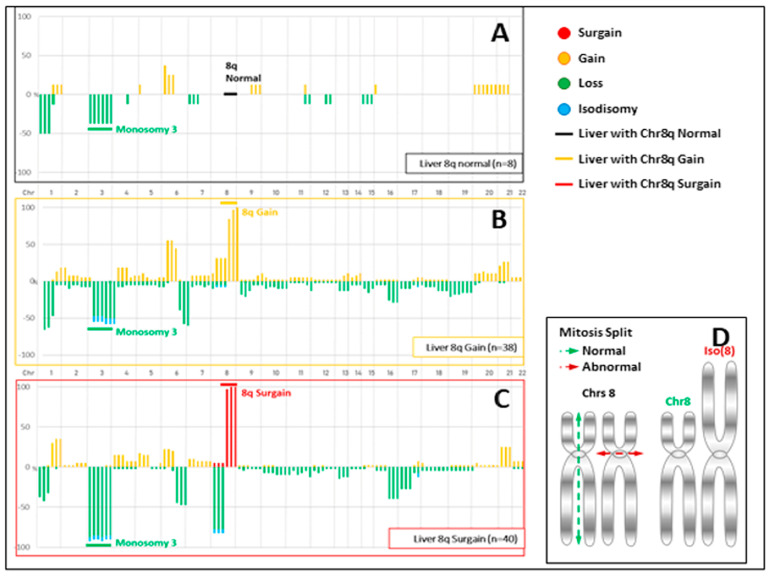
Copy number variation (CNV) profiles of liver metastases divided into 3 groups according to their chromosome 8q (chr8q) status. (**A**) A total of 8 patients with no aberration in chr 8q and monosomy 3 occurring in 37.5% of cases. (**B**) A total of 38 patients with a gain (total or partial) of a unique additional copy of chr 8q (a total of 3 copies of chr8q in a diploid context) and monosomy 3 in 55% of cases. (**C**) A total of 40 patients with extra copies of chr8q (more than 5 and sometimes 7) and monosomy 3 in 90% of cases. (**D**) Abnormal mitosis split mechanisms: As they were mostly associated with a loss of heterozygosity (LOH) of chr8p, either by deletion or isodisomy, the hypothesis of an isochromosome 8 can be proposed. Isochromosomes can be created during mitosis through a mis-division of the centromere. The resulting chromosome contains duplicated arms, which are mirror images. In our model, this iso(8q) could have been reduplicated one or two times, explaining these extra (5 or 7) chromosome 8q copies.

**Figure 3 cancers-16-03407-f003:**
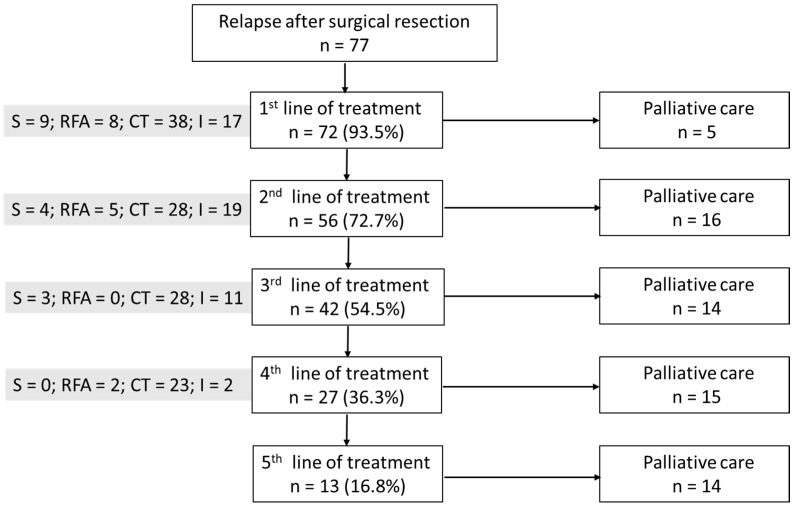
Postoperative treatment after relapse. n: patient number; S: surgical treatment; RFA: radiofrequency ablation; CT: chemotherapy; I: immunotherapy.

**Table 1 cancers-16-03407-t001:** Baseline characteristics: 8q surgain was defined as more than 3 chromosome 8q copies. Four patients had missing data for ciliary body extension. UM: uveal melanoma; AJCC: American Joint Committee on Cancer; LDH: lactate dehydrogenase; MRI: magnetic resonance imaging; ULN: upper limit of normal.

Variables		All (n = 86)	Chromosome 8q Surgain		*p*-Value
			Absent (n = 46)	Present (n = 40)	
Age (years)	≤50	36 (42%)	22 (48%)	14 (35%)	0.23
	>50	50 (58%)	24 (52%)	26 (65%)	
Gender	Female	46 (53%)	24 (52%)	22 (55%)	0.79
	Male	40 (47%)	22 (48%)	18 (45%)	
Ocular UM thickness (mm)	≤20	81 (94%)	43 (93%)	38 (95%)	0.76
	>20	5 (6%)	3 (7%)	2 (5%)	
Ocular UM largest basal diameter (mm)	<10	56 (65%)	27 (59%)	29 (73%)	0.18
	>10	30 (35%)	19 (41%)	11 (28%)	
AJCC tumor category	T1/2/3	52 (60%)	29 (63%)	23 (58%)	0.60
	T4	34 (40%)	17 (37%)	17 (43%)	
Ciliary body extension	Absent	55 (68%)	32 (74%)	23 (61%)	0.18
	Present	26 (32%)	11 (26%)	15 (39%)	
Treatment of ocular UM	I-Disk or Proton beam irradiation	50 (58%)	29 (63%)	21 (53%)	0.32
	Enucleation	36 (42%)	17 (37%)	19 (48%)	
Performance status	0	84 (98%)	46 (100%)	38 (95%)	0.12
	1	2 (2%)	0 (0%)	2 (5%)	
Disease-free interval (months)	≤24	32 (37%)	9 (20%)	23 (58%)	<0.001
	>24	54 (63%)	37 (80%)	17 (43%)	
LDH level	≤ULN	34 (92%)	20 (100%)	14 (82%)	0.05
	>ULN	3 (8%)	0 (0%)	3 (18%)	
Number of liver lesions in MRI	≤2	56 (65%)	30 (65%)	26 (65%)	0.98
	>2	30 (35%)	16 (35%)	14 (35%)	
Largest lesion size in MRI (mm)	≤20	60 (70%)	32 (70%)	28 (70%)	0.97
	>20	26 (30%)	14 (30%)	12 (30%)	
Largest lesion area in MRI (mm^2^)	≤250	46 (53%)	24 (52%)	22 (55%)	0.79
	>250	40 (47%)	22 (48%)	18 (45%)	
Type of liver surgery	Coelioscopy	2 (2%)	1 (2%)	1 (3%)	0.92
	Laparotomy	84 (98%)	45 (98%)	39 (98%)	
Capsular miliary disease	Absent	40 (47%)	26 (57%)	14 (35%)	0.05
	Present	46 (53%)	20 (43%)	26 (65%)	
Parenchymal miliary disease	Absent	70 (81%)	40 (87%)	30 (75%)	0.16
	Present	16 (19%)	6 (13%)	10 (25%)	
Major hepatectomy	Absent	48 (56%)	26 (57%)	22 (55%)	0.89
	Present	38 (44%)	20 (43%)	18 (45%)	
Macroscopically resection	Complete (R0/R1)	80 (93%)	44 (96%)	36 (90%)	0.31
	Incomplete (R2)	6 (7%)	2 (4%)	4 (10%)	
Bleeding loss (mL)	>100	46 (53%)	21 (46%)	25 (63%)	0.12
	≤100	40 (47%)	25 (54%)	15 (38%)	

**Table 2 cancers-16-03407-t002:** Pathological and genetic features of liver metastases: 8q surgain was defined as more than 3 chromosome 8q copies. Fourteen patients had missing data for *CYSLTR2* mutation. The Cassoux classification was defined as follows: low risk—D3/8nl; intermediate risk—M3/8nl or D3/8g; or high risk—M3/8g. The TCGA classification was defined as follows: low risk—D3/8nl; intermediate risk—D3/8g; high risk—M3/8g; or very high risk—M3/8g with 8q surgain. M3/8nl alterations were classified as being of intermediate risk. M3: monosomy 3; D3: disomy 3; 8nl: chromosome 8 normal; 8g: 8q gain; 8q surgain of >3 copies.

Variables		All (n = 86)	Chromosome 8q Surgain		*p*-Value
			Absent (n = 46)	Present (n = 40)	
Liver histopathology	Fusiform	21 (24%)	18 (39%)	3 (8%)	<0.001
	Epithelioid/mixed	65 (76%)	28 (61%)	37 (93%)	
*GNAQ* mutation	Present	43 (50%)	19 (41%)	24 (60%)	0.08
	Absent	43 (50%)	27 (59%)	16 (40%)	
*GNA11* mutation	Present	39 (45%)	23 (50%)	16 (40%)	0.35
	Absent	47 (55%)	23 (50%)	24 (60%)	
*CYSLTR2* mutation	Present	1 (1%)	1 (3%)	0 (0%)	0.33
	Absent	71 (99%)	36 (97%)	35 (100%)	
*SF3B1* mutation	Present	23 (27%)	20 (43%)	3 (8%)	<0.001
	Absent	63 (73%)	26 (57%)	37 (93%)	
*BAP1* mutation	Present	51 (59%)	20 (43%)	31 (78%)	0.001
	Absent	35 (41%)	26 (57%)	9 (23%)	
*EIF1AX* mutation	Present	7 (8%)	6 (13%)	1 (3%)	0.07
	Absent	79 (92%)	40 (87%)	39 (98%)	
Cassoux classification	Low risk	5 (6%)	5 (11%)	0 (0%)	<0.001
	Intermediate risk	23 (27%)	20 (43%)	3 (8%)	
	High risk	58 (67%)	21 (46%)	37 (93%)	
TCGA classification	Low risk	5 (6%)	5 (11%)	0 (0%)	<0.001
	Intermediate risk	23 (27%)	20 (43%)	3 (8%)	
	High risk	21 (24%)	21 (46%)	0 (0%)	
	Very high risk	37 (43%)	0 (0%)	37 (93%)	

**Table 3 cancers-16-03407-t003:** Multivariable analysis of recurrence-free survival (RFS). Multivariable analysis was undertaken by entering all variables associated with RFS at the *p* < 0.05 level in the univariate analysis and then applying a backward stepwise approach to retain significant factors at the *p* < 0.05 level in the final model.

Variable	Multivariable Analysis	
	HR (95% CI)	*p*-Value
Age (years), >50 vs. ≤50	-	-
Ciliary body extension, present vs. absent	-	-
Disease-free interval (months), ≤24 vs. >24	2.2 (1.2–3.8)	0.007
Capsular miliary disease, present vs. absent	-	-
*BAP1* mutation, present vs. absent	-	-
Cassoux classification, high vs. low	-	-
intermediate vs. low	-	-
TCGA classification, very high vs. high	-	-
intermediate vs. high	-	-
low vs. high	-	-
Chromosome 8q surgain, present vs. absent	2.2 (1.3–3.7)	0.005

HR: hazard ratio; CI: confidence interval.

**Table 4 cancers-16-03407-t004:** Multivariable analysis of overall survival (OS). Multivariable analysis was undertaken entering all variables associated with OS at the *p* < 0.05 level in univariate analysis and then applying a backward stepwise approach to retain significant factors at the *p* < 0.05 level in the final model.

Variable	Multivariable Analysis	
	HR (95% CI)	*p*-Value
Age (years), >50 vs. ≤50	-	-
Ciliary body extension, present vs. absent	-	-
Disease-free interval (months), ≤24 vs. >24	2.7 (1.5–4.7)	<0.001
Capsular miliary disease, present vs. absent	-	-
*BAP1* mutation, present vs. absent	-	-
Cassoux classification, high vs. low	-	-
intermediate vs. low	-	-
TCGA classification, very high vs. high	-	-
intermediate vs. high	-	-
low vs. high	-	-
Chromosome 8q surgain, present vs. absent	2.9 (1.6–5.2)	<0.001

HR: hazard ratio; CI: confidence interval.

## Data Availability

The data generated in this study are available upon request from the corresponding author.

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
