# Peer review of "A Clinico-Genetic Score Incorporating Disease-Free Intervals and Chromosome 8q Copy Numbers: A Novel Prognostic Marker for Recurrence and Survival Following Liver Resection in Patients with Liver Metastases of Uveal Melanoma"

_cancers, 2024, doi:10.3390/cancers16193407_

Round 1

Reviewer 1 Report

Comments and Suggestions for Authors

Overall Evaluation

The purpose of this study is to investigate whether the genetic profile of metastases, in conjunction with established prognostic factors, can predict survival outcomes in patients who have undergone surgical resection for liver metastases from uveal melanoma (LMUM) and had genetic analysis of their metastases performed. Through a retrospective analysis of clinical and genetic data from 86 patients who underwent surgical resection for liver metastases, this study confirms, via multivariate analysis, that DFI (Disease-Free Interval) and chromosome 8q surgain (>3 copies) are independent predictors of RFS (Recurrence-Free Survival) and OS (Overall Survival). Based on these findings, a risk scoring system incorporating both clinical characteristics and genetic information was developed, categorizing patients into three risk groups: low, intermediate, and high. The study concludes that a DFI of ≤24 months combined with chromosome 8q surgain identifies patients at high risk of early recurrence and may guide the consideration of peri-operative treatment strategies. While similar reports on predictive markers for RFS and OS following surgery for liver metastases from melanoma have been published (e.g. DOI: 10.1245/s10434-022-12368-5), the relationship between these markers and genetic analysis has not yet been explored, lending this study a certain degree of novelty.

Recommendations

Introduction

In line 44, the authors mentioned “Despite effective treatment of ocular UM, between 30%-50% of patients will present isolated liver metastases (90% of cases)”. It would be helpful to clarify the meaning of "90% of cases" in this context to avoid any potential confusion.

Material and Methods

1.      Could the authors provide reasoning for choosing the Cassoux and TCGA classifications? Are there other classifications that could have been considered?

2.      In line 119, it seems a comma might be missing. A review of the sentence structure would improve readability.

Results

1.      In line 140, the authors state that “A total of 80 patients had a complete resection - 71 without microscopic involvement of the margins and 9 patients with microscopic involvement -. Meanwhile, 6 patients had an incomplete resection”. It would be beneficial to clarify what is meant by "incomplete resection". Additionally, how many cases achieved R0 resection? Should those that did not achieve R0 resection be excluded from certain analyses?

2.      Given that age is related to mRFS, could the authors explain why the age cutoff was set at 50 years?

3.      What is the rationale for setting DFI at 24 months?

4.      To enhance readability, it would be helpful if the authors could explain why three models were chosen for the analysis.

Discussion

In line 247, the author mentions "the first to report". This claim may need reconsideration as the relationship between chromosome 8q surgain and prognosis in LMUM has already been discussed in the article (DOI: 10.1245/s10434-022-12368-5). It would be advisable for the author to review the manuscript to confirm whether this is indeed the first report.

Author Response

Comment 1

Introduction

In line 44, the authors mentioned “Despite effective treatment of ocular UM, between 30%-50% of patients will present isolated liver metastases (90% of cases)”. It would be helpful to clarify the meaning of "90% of cases" in this context to avoid any potential confusion.

Response 1: thank you for your comment. For a better understanding we have corrected the manuscript by deleting ‘90% of cases’.

We modified Introduction Page 2 lines 45: « between 30%-50% of patients will present most of the time isolated liver metastases ».

Comment 2

Material and Methods

  1. Could the authors provide reasoning for choosing the Cassoux and TCGA classifications? Are there other classifications that could have been considered?

Response 2.1 : we are agree with this comment. There are other classifications based on changes on DNA copy number variation (CNVs) or gene expression profilling (GEP).

With regard to CNV, a recent publication (DOI:10.1016/j.xops.2022.100121) shows that, apart from monosomy of chromosome 3 and gain of 8q, abnormalities affecting chromosomes 8p, 1p and 16q could be used to define primary occular tumours with an "ultra-high" risk of metastasis. However, this recent classification has not yet been validated on large series for primary occular tumours and has never been studied in the metastatic setting.

With regard to classification based on alterations in gene expression (GEP), this technique is mainly used in the United States and Canada. We do not use this technique at our centre.

We have therefore decided to use previously published classifications including alterations of chromosomes 3 and 8 without (Cassoux) or with (TCGA) determination of 8q surgain (defined as more than 3 copies).

To clarify our choice, we have modified the manuscript in Material and methods Page 3 lines 94-96 : « Metastases were classified according to two classifications used to establish the prognosis of primary ocular tumours. Those classifications are based on chromosome copy number variation including the status of chromosome 3 and chromosome 8 without (Cassoux) or with (TCGA) identification of an 8q surgain (defined as strictly more than 3 copies). »

  1. In line 119, it seems a comma might be missing. A review of the sentence structure would improve readability.

Response 2.2: We have corrected line 118.

 Comment 3

Results

  1. In line 140, the authors state that “A total of 80 patients had a complete resection - 71 without microscopic involvement of the margins and 9 patients with microscopic involvement -. Meanwhile, 6 patients had an incomplete resection”. It would be beneficial to clarify what is meant by "incomplete resection". Additionally, how many cases achieved R0 resection? Should those that did not achieve R0 resection be excluded from certain analyses?

Response 3.1: incomplete resection corresponds to the presence of unresected macroscopic metastatic liver lesions after surgical treatment. This is R2 surgery for 6 patients. R2 patients were excluded from the RFS analysis. 80 patients had a complete resection : 71 patients underwent resection without microscopic invasion of the margins (R0) and 9 patients underwent resection with microscopic invasion of the margins (R1) and were grouped for OS and RFS analysis.

For best understanding, we have modified the manuscript in Results Page 3 line 141-144:

A total of 80 patients had a complete resection : 71 without microscopic involvement of the margins (R0) and 9 patients with microscopic involvement (R1). Meanwhile, 6 patients had an incomplete resection with unresected macroscopic liver metastases (R2). R2 patients were excluded from the RFS analysis.

  1. Given that age is related to mRFS, could the authors explain why the age cutoff was set at 50 years?

Response 3.2 : to our knowledge, there is no established data in the literature on the prognostic value of age in patients with liver metastases from uveal melanoma. We therefore decided to use the median age of our patients cohort.

  1. What is the rationale for setting DFI at 24 months?

Response 3.3 : a DFI value at 24 months was chosen on the basis of data published by 2 different teams (10.1371/journal.pone.0120181 and 10.3390/cancers11060863). In these two nomograms based on metastatic patients, a DFI of less than 24 months had a strong negative prognostic value.

  1. To enhance readability, it would be helpful if the authors could explain why three models were chosen for the analysis.

Response 3.4 : the majority of metastatic patients have monosomy 3. The TCGA classification, which is now used by many teams, has shown that there is a prognostic difference in relation to the number of copies of chromosome 8q in patients with monosomy 3. This classification distinguishes between high-risk patients (M3, gain of 8q ≤3 copies) and very high-risk patients (M3, 8q surgain > 3 copies). In this study we wanted to test the prognostic value of 8q surgain in metastatic patients undergoing surgery.

We therefore compared three models : the Cassoux classification (model 1) without 8q surgain information, the TCGA classification (model 2) with 8q surgain information and 8q surgain alone (model 3).

To enhance readability, we have modified the manuscript in Material and methods Page 3 line 122-126:

« To test the prognostic value of 8q surgain in metastatic patients undergoing surgery, three Cox multivariable models were assessed, each one exploring different genetic classifications : the Cassoux classification (model 1[17]) without information about 8q surgain ; the TCGA classification (model 2 [18]) with information about 8q surgain and 8q surgain alone without status of chromosome 3 (model 3). »

Comment 4 

Discussion

In line 247, the author mentions "the first to report". This claim may need reconsideration as the relationship between chromosome 8q surgain and prognosis in LMUM has already been discussed in the article (DOI: 10.1245/s10434-022-12368-5). It would be advisable for the author to review the manuscript to confirm whether this is indeed the first report.

Response 4 : we thank the reviewer for this comment. In the study (DOI: 10.1245/s10434-022-12368-5) the genetic classification of metastases was  a statistically significant factor for DFS and OS in univariate analysis but not in multivariate analysis. The genetic characteristics of metastases therefore had no prognostic impact on survival in this study.  Furthermore, in this study, the genetic classification of metastases did not include data on 8q copy number (normal 8q or 8q gain).

This is why we have written "This retrospective study is the first to report the adverse prognostic impact of chromosome 8q surgain in patients with LMUM "

Reviewer 2 Report

Comments and Suggestions for Authors

The manuscript entitled “A clinico-genetic score incorporating disease-free interval and chromosome 8q copy number: a novel prognostic marker for recurrence and survival following liver resection in patients with liver metastases of uveal melanoma” reports on the association of disease free interval (DFI) and chromosome 8q status with recurrence and survival in patients treated by liver resection. The study is well thought out, performed and described, the results are correctly interpreted and discussed. The paper is well written despite some typos.

Comments:

A pro-metastatic gene on chromosome 8q has already been detected (DDEF1 is located in an amplified region of chromosome 8q and is overexpressed in uveal melanoma. Ehlers JP, Worley L, Onken MD, Harbour JW. Clin Cancer Res. 2005 May 15;11(10):3609-13. doi: 10.1158/1078-0432.CCR-04-1941).

Given the independence of DFI and chr8q status the question arises of whether cases with short DFI are driven by unknown factors other than chr8q gain. The authors should therefore show the chr8q copy number in the two classes of DFI (<24 and >24 months).

The authors use “gain” and “surgain”. If they intend there to be a difference this should be stated, if not, “gain” should be used instead of “surgain”.

The association of GNAQ vs. GNA11 mutations with relapse should be shown since other reports in addition to the one cited show a higher metastatic potential for GNA11.

Row 263 “mainly” should read “main”

Row 360 “concider” should read “consider”

Comments on the Quality of English Language

Row 263 “mainly” should read “main”

Row 360 “concider” should read “consider”

There might be other typos, please check carefully

Round 2

Reviewer 1 Report

Comments and Suggestions for Authors

In the response letter, the authors mention that 9 patients underwent resection with microscopic involvement (R1). The question remains: why were these patients not excluded from the statistical analysis?
Additionally, in the manuscript, the authors should cite the two references that provide the rationale for defining DFI at 24 months.

Round 3

Reviewer 1 Report

Comments and Suggestions for Authors

no comments